# Classification and analysis of text transcription from Thai depression assessment tasks among patients with depression

Adirek Munthuli[1,2], Pakinee Pooprasert[2], Nittayapa Klangpornkun[1,2], Phongphan Phienphanich[1,2], Chutamanee Onsuwan[2,3], Kankamol Jaisin[4], Keerati Pattanaseri ![ORCID][4], Juthawadee Lortrakul ![ORCID][4] *, Charturong Tantibundhit[1,2]

1 Department of Electrical and Computer Engineering, Thammasat School of Engineering, Thammasat University (Rangsit Campus), Khlong Luang, Pathum Thani, Thailand, 2 Center of Excellence in Intelligent Informatics, Speech, and Language Technology, and Service Innovation (CILS), Thammasat University, Khlong Luang, Pathum Thani, Thailand, 3 Department of Linguistics, Faculty of Liberal Arts, Thammasat University (Rangsit Campus), Pathum Thani, Thailand, 4 Department of Psychiatry, Faculty of Medicine Siriraj Hospital, Mahidol University, Bangkok Noi, Bangkok, Thailand

* juthawadee.lor@mahidol.ac.th

**Data Availability Statement:** All relevant data are within the paper and its Supporting information files.

## Abstract

Depression is a serious mental health disorder that poses a major public health concern in Thailand and have a profound impact on individuals' physical and mental health. In addition, the lack of number to mental health services and limited number of psychiatrists in Thailand make depression particularly challenging to diagnose and treat, leaving many individuals with the condition untreated. Recent studies have explored the use of natural language processing to enable access to the classification of depression, particularly with a trend toward transfer learning from pre-trained language model. In this study, we attempted to evaluate the effectiveness of using XLM-RoBERTa, a pre-trained multi-lingual language model supporting the Thai language, for the classification of depression from a limited set of text transcripts from speech responses. Twelve Thai depression assessment questions were developed to collect text transcripts of speech responses to be used with XLM-RoBERTa in transfer learning. The results of transfer learning with text transcription from speech responses of 80 participants (40 with depression and 40 normal control) showed that when only one question ($Q_1$) of "How are you these days?" was used, the recall, precision, specificity, and accuracy were 82.5%, 84.65, 85.00, and 83.75%, respectively. When utilizing the first three questions from Thai depression assessment tasks ($Q_1 - Q_3$), the values increased to 87.50%, 92.11%, 92.50%, and 90.00%, respectively. The local interpretable model explanations were analyzed to determine which words contributed the most to the model's word cloud visualization. Our findings were consistent with previously published literature and provide similar explanation for clinical settings. It was discovered that the classification model for individuals with depression relied heavily on negative terms such as 'not,' 'sad,', 'mood', 'suicide', 'bad', and 'bore' whereas normal control participants used neutral to positive terms such as 'recently,' 'fine,', 'normally', 'work', and 'working'. The findings of the study suggest that screening for depression can be facilitated by eliciting just three

**Funding:** This research project is supported by National Research Council of Thailand (NRCT): NRCT5-RGJ63008-102, as a part of the Adirek Munthuli's PhD scholarship program. This study was also supported by National Research Council of Thailand in the form of a grant (Grant No. 2564N00632250) awarded to Charturong Tantibundhit.

**Competing interests:** The authors have declared that no competing interests exist.

questions from patients with depression, making the process more accessible and less time-consuming while reducing the already huge burden on healthcare workers.

## Introduction

Depression is one of the most prevalent forms of mental illness, affecting approximately 280 million people worldwide [1, 2]. In the first year of the global coronavirus disease 2019 pandemic, the prevalence of depression reached an all-time high of approximately 28%, and has risen steadily over the following decades [1, 3]. Additionally, social media can also have an impact on increasing the prevalence of depression, especially the potential conflict between Russia and Ukraine in 2022. Individuals may have an increased likelihood of developing depressive symptoms [4]. Depression not only poses a risk to the individual but is also a major contributor to the overall economic burden and is responsible for more than half a million suicides annually [5]. In the United States in 2018, depression ranked third in terms of economic impact just behind cardiovascular disease and cancer, with an estimated cost of 326.2 billion dollars [6].

Thailand is ranked fourth in Southeast Asia for the number of reported cases of depressive disorders, totaling 2.9 million people, or 4.4% of the population [7]. Despite the fact that 52% of Thailand's people live in urban areas, which provide a facility for support in the event of depression, there are only about 882 psychiatrists in the country, which is insufficient to assist the entire population [8, 9]. Furthermore, there have been numerous reports which indicated that a significant number of people in rural areas are underdiagnosed with depression [10, 11]. A study reported that the prevalence of depression in the elderly in rural areas was as high as 18.5% [10]. Another study revealed that the prevalence of depression among Thai hill tribe residents is approximately 39.1% [11].

These statistics demonstrate the increasingly burdensome nature of depression amongst the Thai population, and stresses the importance of developing a supportive framework to help battle this significant mental health disease [7–11]. As a result, the Department of Mental Health, Ministry of Public Health is developing a strategy to provide more access to the nation's medical and mental health infrastructure [7, 12, 13]. This program will be carried out by public health department representatives using depression assessment tools such as Thai version of the Patient Health Questionnaire (PHQ-9) [14] and the Thai version of the Hamilton Depression Rating Scale (HDRS or HAM-D) [15]. As a result, the percentage of people with access to depression services at health facilities across the Thai nation increased to 86.73% (out of 1,356,736 inpatients) [16]. While these efforts have produced somewhat favorable results, it also poses a significant burden for medical personnel and is an arguably unrealistic and unsustainable approach [9, 17]. Thus, the integration of Artificial Intelligence (AI) technology may help solve these shortcomings and provide a more innovative and sustainable solution to help fight the crisis of depression in Thailand.

AI is one of the most cutting-edge scientific developments and has led to paradigm shifts in the way diseases are screened, diagnosed, and treated [18]. Various studies have utilized machine learning (ML) or deep learning (DL) frameworks to extract and learn multi-scale feature representations from unimodal or multimodal data in order to classify depression or predict depressive severity [19–25]. Text is a promising modality for screening depression due to its non-invasiveness, cost-effectiveness, and ability to accumulate records for tracking mental health over a given period. Various types of texts can be leveraged with the help of AI for

depression screening, such as social media posts, handwritten notes (either from individuals or discharge summaries from medical professional), and written text transcripts of speech [22–31].

The majority of recent studies have examined the potential for detecting depressive symptoms using informal text of social media posts [22–26]. A study from Wani et al. (2022), is among the existing literature that used deep learning, such as convolutional neural network (CNN) and a long-short term memory (LSTM) for depression classification from around 53,000 of social media posts in English language and produced state-of-the-art results with classification accuracy of 99.02% [24]. However, CNN or LSTM based language models require enormous datasets to accommodate a vast number of parameters, which is a significant drawback. This makes it simple to overfit the model and difficult to generalize when there are insufficient data points [32]. To address this issue, there has been a growing interest in leveraging "data-centric AI movement" for depression classification [25–31].

The "data-centric AI movement" has made it possible for small data solutions to solve the most significant AI problems. Only data of high quality may be adequate to explain an AI system [33]. Transfer learning from a large model or Transformer-based Pre-trained Language Models (T-PTLM), such as a Google's Bidirectional Encoder Representations from Transformers (BERT) [34] or a Facebook's Cross-lingual Language Model (XLM) [35] and a Robustly Optimized BERT Pretraining Approach (RoBERTa) [36], enables the utilization of multiple benchmarks with minimal task-specific tuning [37, 38]. The most notable feature of these pre-trained models is that they were trained using massive amounts of data [34–36, 38]. For instance, Google's BERT model was trained on 16 GB data (approximately 3.3 billion words) and contains approximately 110–336 M parameters [34], and Facebook's RoBERTa was trained on 160 GB data (10x more than Google's BERT) [36].

In recent years, several studies have developed classification models from social media posts using vanilla fine-tuning of BERT and the model can successfully achieve up to 98% accuracy [25–27, 30, 31]. Instead of using social media posts, Dai et al. (2021) provided more examples of their work. They utilized transfer learning from four variants of the BERT model to identify depression in discharge summaries, and RoBERTa yielded the highest classification F1-score of 0.830 [28]. Another example was the work by Senn et al. (2022), which used transcripts of 12 clinical interview questions and a transfer learning approach with three variants of BERT and an ensemble of extra LSTM and attention layers. The results indicated that RoBERTa tended to perform better with basic architectures individually without ensemble. However, the best result was achieved using the BERT architecture with ensemble strategy of only one question, with an F1-score of 0.93. When using all the questionnaires, the F1-score dropped to $0.62 \pm 0.12$ (mean ± S.D.) [29].

There are an infinite number of ways to build an accurate classification model, depending on the type of studies, model architectures, datasets, etc., making it difficult to pinpoint for the best fit model in clinical application [19–31]. Few studies attempted to explain what happens within the AI model in order to apply this knowledge in clinical settings [39, 40]. Explainable Artificial Intelligence (XAI) is a technique used to understand the decisions or predictions made by AI. Due to XAI's emphasis on making AI systems understandable, practitioners and end-users can comprehend or get a sense of what the systems are thinking [39, 40]. Additionally, the information derived from XAI can make the classification model's more reliability, and trustworthiness, in addition to helping meet the legal and ethical obligations associated with the development and deployment of AI models in clinical settings [39]. There are currently ongoing studies that apply XAI to examine text transcripts from dementia patients utilizing transformers network [41]. In additional studies, these strategies were also applied to depressed patients in Norwegian languages on social media posts [42].

Most current research has attempted to study text from social media, as it is widely available and can provide a large amount of data for analysis, as well as a window into an individual's thoughts and feelings. Even though a significant number of social media texts can be mined for training, this method has limitations [33]. Since the majority of information is classified according to the hashtag or content of the posts, it is particularly problematic whether posts are originated from depressed users or are obviously fabricated. This makes the model training on a non-clinically proven dataset [22–26, 30, 31]. In contrast, text transcripts from spoken responses are better suited for classifying depression. They provide a more in-depth view of the user's emotional state, as they are often more detailed and provide more contexts and nuances.

Due to the above limitations and the trend towards data-centric AI, the purpose of this study is to develop a depression classification model for text transcript data obtained from twelve questions from Thai depression assessment tasks [43] completed by 80 participants under clinical settings. The selected model for this study is XLM-RoBERTa$_{BASE}$, a transformer-encoder model that supports multiple languages, including Thai [44], due to the support provided by various studies for the best performance of RoBERTa architectures in depression screening [28–30]. Moreover, Local Interpretable Model-agnostic Explanations (LIME) [40] will be used to visualize the fine-tuned XLM-RoBERTa model and determine which words from the text transcripts are the most critical for Thai depression screening. Our four main research contributions include:

1. Fine-tuning XLM-ROBERTA$_{BASE}$ [44] to text transcripts elicited from spoken responses to twelve questions of Thai depression assessment tasks [43].

2. Assessing the effectiveness of Thai depression assessment tasks for detecting depression from text responses in speech and outline a set of questions that yield the most accurate results in clinical settings.

3. Explaining a black-box model from fine-tuned XLM-RoBERTa using weight explained from LIME explanations [40].

4. Visualizing weights from LIME explanations [40] in a word cloud format and explaining their relevance to clinical settings.

This research has important clinical implications, as it demonstrates that a highly accurate model for classifying depression can be produced using a limited set of high-quality text transcripts from speech in clinical settings. This is particularly noteworthy for uncommon domestic languages such as Thai, where data availability is often a significant issue. Additionally, the explanations provided by LIME through the black-box model can be interpreted and understood in a clinical context.

## Materials and methods

The goal of this study is to examine the potential of using a limited-set of text transcripts from speech responses to fine-tune a T-PTLM to create a promising classification model [38]. The main issues addressed in this research are (1) the scarcity of textual transcripts from speech collected in clinical settings, (2) how to develop a highly accurate model using T-PTLM with a limited dataset, and (3) how the results can be applied and trusted in clinical settings. Fig 1 illustrates the process from data collection, data preparation, and fine-tuning of the model. The details of this process are provided in the following subsections.

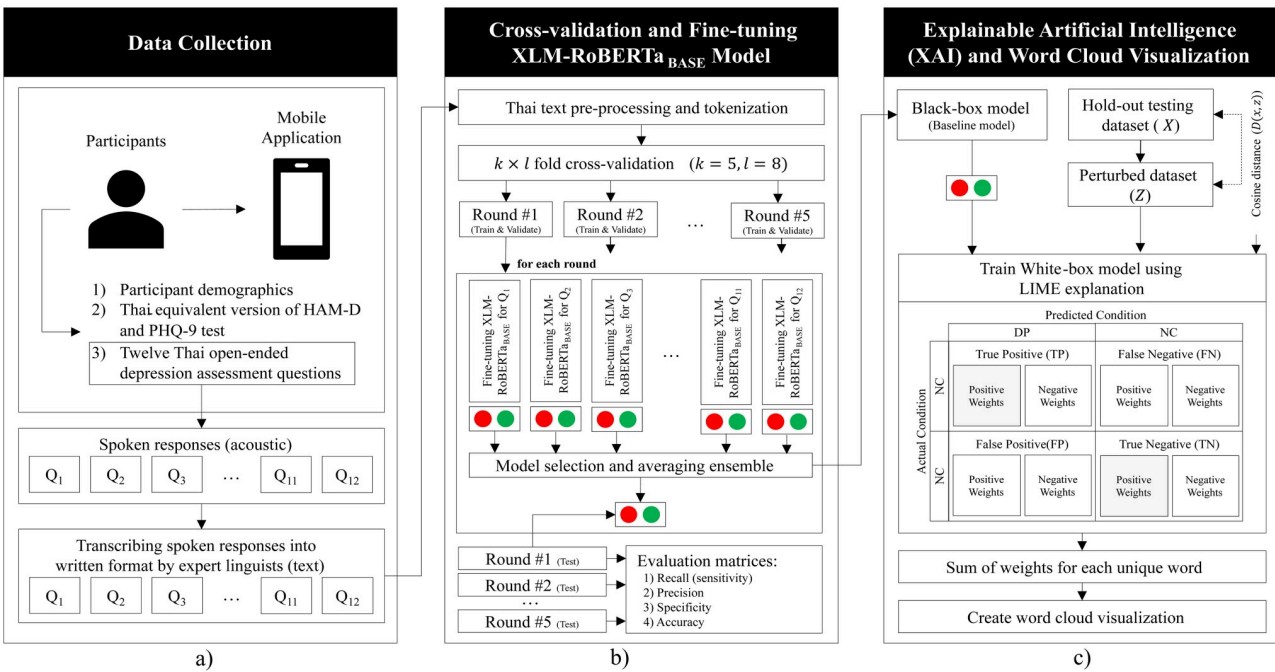

**Fig 1. A flow diagram of the article's process including a) data collection and speech elicitation, which are then transformed into text transcriptions; b) text pre-processing and normalization, followed by data preparation for cross-validation in transfer learning to the model for each question to compute evaluation matrices; c) training of the model for LIME explanation and utilization of the weights from the explanation for creating word clouds for both the normal control group and the depression group.**

## Research instruments

The research instruments utilized in this study consist of three essential parts: 1) Information about the participant demographics, including participant group, gender, and age 2) Thai equivalent version of the HAM-D and PHQ-9 tests [14, 15] 3) Thai open-ended depression assessment tasks comprised of a set of 12 questions divided into two parts: six questions for reviews of symptom ($Q_1 - Q_6$) and six questions for describing emotional experiences ($Q_7 - Q_{12}$), as shown in Table 1.

## Study design and participants

This study was performed in accordance with the Faculty of Medicine Siriraj Hospital, Mahidol University, and approval was obtained from the Siriraj Institution Review Board (IRB) (ID number: 732/2563). There were a total of 80 participants, as shown in more detail in Table 2. Of these, 40 participants were given a depression diagnosis, and the other 40 participants served as normal control. Inclusion criteria includes: participants who are over 18 years of age and speak Central Thai Dialect. Depressed patients must be diagnosed with Major Depression, Bipolar Disorder Depressive Episode, or Persistent Depressive Disorders (Dysthymia), according to the Diagnostic and Statistical Manual of Mental Disorders, DSM-5 diagnostic criteria [45] and their Thai equivalent version of PHQ-9 and HAM-D scores must met the criteria of depression (PHQ-9 score $\geq$ 9 and HAM-D score $\geq$ 8) [14, 15]. The severity of depression was classified based on Thai equivalent version of HAM-D score as gold standard diagnosis [15].

**Table 1. The developed thai depression assessment tasks [43].**

| Variables | Items | Questions |
|---|---|---|
| **Review of Symptoms** | | |
| • Clinical follow-up | Q1 | How are you these days? |
| | Q2 | What do you think your symptoms are? |
| • Neutral state of mentality | Q3 | Please describe your feelings in the past two weeks. |
| | Q4 | How have you felt change from before? |
| • Perception of life problems | Q5 | How is your life now? |
| | Q6 | Tell me about things that strengthen your life. |
| **Emotional experiences** | | |
| • Neutral-emotion induction | Q7 | What is your life goal? |
| | Q8 | If you have free time of a year, what would you like to do? |
| • Positive-emotion induction | Q9 | Talk about an event that impresses you. |
| | Q10 | Talk about something you enjoy doing or doing that makes you happy. |
| • Negative-emotion induction | Q11 | Tell us about the biggest obstacles you've ever encountered in your life. |
| | Q12 | Talk about what makes you feel afraid. |

## Data collection procedure

A prospective matched case-control design was utilized in this study. The primary objective of this study was to classify depression via text transcribed from speech responses of twelve questions from the Thai depression assessment tasks out of a full set of 32 questions described in previous study [43]. A mobile application version of the Thai psychiatric test kit with three main components described in the preceding section was developed [43]. Psychologists on the

**Table 2. Participants' demographic data.**

| Participant Type | Depression Severity (Thai HAM-D) | | | | Normal Control |
|---|---|---|---|---|---|
| | **Mild** | **Moderate** | **Severe** | **Total** | **Total** |
| No Depression | - | - | - | - | 40 |
| Unipolar Depression | 9 | 11 | 13 | 33 | - |
| Bipolar Type I | 1 | 2 | - | 3 | - |
| Persistent | 1 | - | 3 | 4 | - |
| **Gender** | | | | | |
| Male | 1 | 4 | 5 | 10 | 9 |
| Female | 10 | 9 | 11 | 30 | 31 |
| **Thai PHQ-9** | | | | | |
| No Depression | 2 | 1 | - | 3 | 40 |
| Depression | 9 | 12 | 16 | 37 | - |
| **Age (Range)** | **21–63** | **23–63** | **23–68** | **21–68** | **20–65** |
| ≤ 30 | 7 | 6 | 5 | 18 | 24 |
| 31–40 | - | 3 | 6 | 9 | 5 |
| 41–50 | 2 | 1 | 1 | 4 | 5 |
| 51–60 | 1 | - | 3 | 4 | 4 |
| ≥ 60 | 1 | 3 | 1 | 5 | 2 |
| Avg. ± SD | 35.09 | 36.32 | 38.13 | 36.80 | 34.00 |
| Std. Dev. | 13.95 | 15.52 | 14.00 | 14.18 | 12.36 |

| | |
|---|---|
| **Moderate Depression** Female, Age = 26 Thai HAM-D Score = 16 Thai PHQ-9 Score = 23 | **[Thai]** "ช่วงนี้ก็จะ *เอ่อ* หลังจากการรักษาก็ดีขึ้นมาบ้างแล้วค่ะ แต่ว่าอาการหลัก ๆ ก็ยังมีอยู่ อย่างเช่น เบื่อ แล้วก็ *เอ่อ* กระสับกระส่าย นอนไม่หลับค่ะ *เอิ่ม* สมาธิในการเรียนก็ลดลงค่ะ แล้วก็ รู้สึกว่าไม่อยาก ไม่อยากจะแบบตื่นไปเรียนหรือทำสิ่งที่เคยทำ ไม่อยาก *เอ่อ* ไม่อยากอ่านหนังสือ ไม่อยากทำการบ้าน ไม่อยากพบปะผู้คน ประมาณนี้ค่ะ" **[English Translation]** "Now it's *uh*, after the treatment it's a bit better. But the main symptoms are still there, for example, bored and *uh*, restless, can't sleep, *uh*, lowered concentration in studies, and I don't want to.. I don't want to be like waking up to school or doing what I used to do. Don't want, *uh*, don't want to read books, don't want to do homework, don't want to meet people this time. |
| **Severe Depression** Male, Age = 30 Thai HAM-D Score = 24 Thai PHQ-9 Score = 20 | **[Thai]** "ช่วงนี้รู้สึกโกรธตัวเองที่บางครั้งบางทีทำไมพูดไม่ตรงกับคนอีกกลุ่มนึงและพูดตรงกับคน อีกกลุ่มนึง จนเป็นสาเหตุให้เกิดความขาดความน่าเชื่อถือและเป็นสิ่งที่ทำให้ตัวเองโกรธกัน โกรธ ภายในตัวเองซ้ำไปซ้ำมาจนทำให้ตัวเองดีเพรส (depress) **และนำไปสู่ภาวะที่คิดไม่อยากอยู่ครับ**" **[English Translation]** "Lately, I feel angry with myself for sometimes saying something wrong with another group of people and speaking directly to another group of people. This causes a lack of credibility, and something that makes me angry. The anger builds, over and over, until it causes me to be depressed **and leads to a state of don't want to live.**" |
| **Normal Control** Female, Age = 30 Thai HAM-D Score = 5 Thai PHQ-9 Score = 1 | **[Thai]** "สบายดีค่ะ ทำงานปกติ หนูมาทำงานตอนเช้า ทำงานปกติ กวาดถู เช็ดกระจก ไม่มีอะไร เปลี่ยนแปลงค่ะ แล้วก็ ร่างกายหนู สุขภาพแข็งแรงดีค่ะ แล้วก็ทำงานสนุก ทำงานสนุก" **[English Translation]** "I'm fine. I work normally. Come to work in the morning. Do the same things as sweeping, mopping, wiping the mirror. Nothing has changed. My body is in good health. And it's quite fun to work - it's fun!" |

**Fig 2. Samples of transcription of speech response from participants with original language (Thai) and English translation.**

research team recorded the subjects' general information, including their gender and age. The participants were subsequently required to complete the Thai version of PHQ-9 self-report questionnaire. Then, psychologists began recording Thai version of HAM-D scores in accordance with clinical practice guidelines [14, 15]. After the Thai versions of PHQ-9 and HAM-D scores have been recorded, the elicitation process begins, with each participant sitting in a quiet room with a smartphone and a tripod to elicit speech data from Thai depression assessment tasks. This procedure was repeated until the participant completed all twelve questions as in Fig 1a).

**Text transcription and labeling.** As in Fig 1a), the elicited speech responses were manually transcribed by six linguists in the research team by dividing the work in equal proportions with the same labeling protocol. Each audio file, was transcribed by a linguist, and the answer was transcribed based on the participants' responses without trimming off any part of the answer. The results may contain English transliteration, hesitation, stuttering, etc., as demonstrated in Fig 2.

**Thai text pre-processing and tokenization.** The text transcript from linguists is pre-processed using the PyThaiNLP [46] library to remove any special characters (e.g. commas, double-spacing, etc.) and normalize to meet the standard Thai writing style. Since the text transcript may include some English words produced by participants, all English words are transliterated into Thai text. The pre-processed transcript is then tokenized using the WangchanBERTa tokenizer [47] before applying it for transfer learning to the XLM-RoBERTa model [44].

**Choice of T-PTLM.** T-PTLM was used to train a massive amount of unlabeled text data and fine-tune small task-specific dataset, which is suitable to our primary objective [38]. More

advantageously, there are many multi-language support for T-PTLM available today, which allows for uncommon domestic languages to gain access to T-PTLM [34, 35, 38, 44]. Some of the most established encoder-based T-PTLMs include mBERT and XLM-RoBERTa, which were developed from the tremendously successful monolingual English models BERT and RoBERTa [34, 44]. However, many studies have raised the limitations of the representations of low-resource languages of mBERT model compared to XLM-RoBERTa model [48]. Due to those limitations, the primary T-PTM model, which was used in this study is based on XLM-RoBERTa$_{BASE}$ model [44].

**Implementation of fine-tuning XLM-RoBERTa model.**  The XLM-RoBERTa model was implemented for fine-tuning the data using the Keras library [49] on an NVIDIA RTX A6000 with 128 GB of RAM and 1 TB of SSD. The model was trained using a triangular learning rate policy [50] with a reduction factor of 2 and a reduction on plateau of 4. Other hyperparameters included a maximum of 50 epochs of training, a batch size of 8, and a learning rate of 1e-5. Furthermore, the model for each question was trained individually to examine the potential of the questions for the performance of the classification model.

**K×L-fold stratified and nested cross validation.**  This study evaluated the model using K×L-fold stratified and nested cross-validation techniques [51]. This is a variation of cross-validation techniques in which all samples were considered as a test dataset. The sample was divided into $k$ subsamples of equal size (folds). The $k − 1$ folds were used to for the training and validating datasets, while the remaining fold was used for the testing dataset. For $k − 1$ folds, each fold is partitioned into $l$ random partitions, with the $l − 1$ partition used for model training and the remaining partitions used for model validation. We used $k = 5$ and $l = 8$ for these experiments, so for each round (of 5 rounds), 70% is used for training, 10% for validating, and 20% for testing the model. Hence all data points (80 participants) were used to evaluate to model by summing up of classification of all five rounds.

**Evaluation matrices.**  Our goal is to maximize recall (sensitivity), precision, and specificity of the model, as these matrices are widely used for clinical applications. Additionally, accuracy is reported to provide a holistic view of performance. Thus, in this work, the performance of a model is evaluated based on a variety of metrics, including recall, precision, specificity, and accuracy, using the following equations [52]:

$$\text{Recall} = \frac{\text{TP}}{\text{TP} + \text{FN}} \tag{1}$$

$$\text{Precision} = \frac{\text{TP}}{\text{TP} + \text{FP}} \tag{2}$$

$$\text{Specificity} = \frac{\text{TN}}{\text{TN} + \text{FP}} \tag{3}$$

$$\text{Accuracy} = \frac{\text{TP} + \text{TN}}{\text{TP} + \text{TN} + \text{FP} + \text{FN}} \tag{4}$$

where TP is true positive (depression predicted as depression), TN is true negative (normal control predicted as normal control), FP is false positive (normal control predicted as depression), and FN is false negative (depression predicted as normal control).

**Model explanation and visualization.**  The model will later be explained by LIME explanation [40] to determine which words contribute the most for depression screening. The idea of LIME explanation was to randomly make an absence of words from text transcripts (perturb) and to create white-box model that resembles the black-box model (fine-tuned

XLM-RoBERTa model) and minimizes the following equations:

$$\arg \min_{g \in G} \mathcal{L}(f, g, \pi_x) + \Omega(g) \tag{5}$$

$$\pi_x(z) = \exp(-D(x, z)^2/\sigma^2) \tag{6}$$

where $x$ is the instance to be explained, $z$ is the perturbed instance, $f$ is the black-box model to be explained, $G$ is the hypothesis space of white-box models, $\mathcal{L}(f, g, \pi_x)$ is the deviation between the black-box model ($f$) and the white-box model ($g$), as weight measured on a perturbed version of the text transcripts $x$, represented by $\pi_x(z)$, $\Omega(g)$ is a regularization term that controls the complexity of the interpretable model ($g$), $D(x, z)$ is a cosine similarity between $x$ and $z$, and $\sigma$ is kernel width.

The goal of LIME is to find the white-box model ($g$) that is closest to the black-box model ($f$) and at the same time has the lowest complexity, as determined by $\Omega(g)$. The result of LIME is an explanation of the predictions of the black-box model ($f$) for the instance $x$. In this work, we implement a LIME analysis using the ELI5 library [53]. With ELI5, we can obtain an interpretable description of the model's predictions along with the individual contributions of each word ($\pi_x(z)$).

Since binary classification models produce four different weights in LIME explanations, namely the weight of the true positive class (depression predicted as depression), the weight of the false negative class (depression predicted as normal control), the weight of the false positive class (normal control predicted as depression), and the weight of the true negative class (normal control predicted as normal control), the relationship between words and the prediction of depression or normal control will be examined by summarizing the LIME weights from the true positive and true negative classes as a word cloud. To calculate this, we summed all the weights that had a positive effect on each class and normalized them to a scale between 0 and 1.

## Results

### Classification results of fine-tuned XLM-RoBERTA

After performing K×L-fold cross-validation to fine-tune the XLM-RoBERTa model with each question, we found that considering only one question $Q_1$, "How are you these days?", the model has a sensitivity of 82.50%, a precision of 84.62%, a specificity of 85.00%, and an accuracy of 83.75%. However, the result matrix gradually decreases for subsequent questions. For $Q_2$, "What do you think your symptoms are?", the sensitivity remains the same at 82.50%, but the precision dropped to 80.49%, the specificity dropped to 80.00%, and the accuracy dropped to 81.25%. Similarly, for $Q_3$, "Please describe your feelings in the past two weeks?", the precision dropped to 78.57%, the specificity dropped to 77.50%, and the accuracy dropped to 80.00%, as seen in Table 3. $Q_1 - Q_6$ were the six out of the twelve questions that yielded promising results, whereas the remaining questions could not be fine-tuned in order to produce favorable outcomes. In other words, the training accuracy of the six questions of $Q_7 - Q_{12}$ has converged, but the testing performance was roughly at 50% as seen in Table 3. Hence, convergence is not possible in this case for $Q_7 - Q_{12}$, which will be discussed later.

### Multiple-questionnaire for averaging ensembles

The findings from the prior section indicated that using only $Q_1$ can result in a sensitivity of 82.50%, a precision of 84.62%, a specificity of 85.00%, and an accuracy of 83.75%, whereas utilizing only $Q_2$ can result in a sensitivity of 82.50%, a precision of 80.49%, a specificity of

**Table 3. Recall, precision, sensitivity, accuracy, and numbers of support (positive predicted) for each group of six questions of reviews of symptom ($Q_1 - Q_6$) and six questions of describing emotional experiences.**

| Evaluation Matrix | $Q_1$ | $Q_2$ | $Q_3$ | $Q_4$ | $Q_5$ | $Q_6$ | Average | S.D. |
|---|---|---|---|---|---|---|---|---|
| Recall (sensitivity) | 82.50 | 82.50 | 82.50 | 80.00 | 72.50 | 72.50 | 78.75 | 4.94 |
| Precision | 84.62 | 80.49 | 78.57 | 72.73 | 74.36 | 74.36 | 77.52 | 4.54 |
| Specificity | 85.00 | 80.00 | 77.50 | 70.00 | 75.00 | 75.00 | 77.08 | 5.10 |
| Accuracy | 83.75 | 81.25 | 80.00 | 75.00 | 73.75 | 73.75 | 77.92 | 4.31 |
| No. Support (DP) | 33 | 33 | 33 | 32 | 29 | 29 | 31.50 | 1.97 |
| No. Support (NC) | 34 | 32 | 31 | 28 | 30 | 30 | 30.83 | 2.04 |
| Evaluation Matrix | $Q_7$ | $Q_8$ | $Q_9$ | $Q_{10}$ | $Q_{11}$ | $Q_{12}$ | Average | S.D. |
| Recall (sensitivity) | 50.00 | 42.50 | 50.00 | 57.50 | 42.50 | 32.50 | 45.83 | 8.61 |
| Precision | 64.52 | 60.71 | 55.56 | 51.11 | 47.22 | 54.17 | 55.55 | 6.29 |
| Specificity | 72.50 | 72.50 | 60.00 | 45.00 | 52.50 | 72.50 | 62.50 | 11.94 |
| Accuracy | 61.25 | 57.50 | 55.50 | 51.25 | 47.50 | 52.50 | 54.25 | 4.87 |
| No. Support (DP) | 20 | 15 | 20 | 23 | 15 | 13 | 17.67 | 3.88 |
| No. Support (NC) | 29 | 29 | 24 | 18 | 21 | 29 | 25.00 | 4.77 |

80.00%, and an accuracy of 81.25%. When all of the questions were examined at once using averaging ensemble, it was found that the model based on the first three questionnaires ($Q_1 - Q_3$) had the potential to attain a sensitivity of 87.50%, a precision of 92.11%, a specificity of 92.50%, and an accuracy of 90.00%. However, when all five questions ($Q_1 - Q_5$) were used, the results produced a sensitivity of 90.00% but the precision dropped to 83.72%, the specificity dropped to 82.50%, and the accuracy dropped to 86.25%.

## Word cloud from LIME explanation

Only word clouds from question $Q_1$ achieved the highest classification matrix results, which are included in this section. In the $Q_1$, the word 'not' provided 10.26% of the total weight for classifying of depression, followed by the word 'sad', which contributed 3.75% of the total weight, and the word 'mood,' which contributed 3.28% of the total weight. The remaining significant words were 'suicide' (1.83%), 'bad' (1.44%), and 'bore' (1.33%), as seen in Fig 3a). The negative terms such as 'stress,' 'disappoint,' and 'hurt' were also included in the word cloud.

When the word cloud of $Q_1$ from the normal control group in Fig 3d) was examined, it was found that the word 'recently' was the one that was most likely to contribute to the model. It contributed 17.39% of the total weight to the cloud. The word 'fine' (10.99%) and 'normally' (5.01%) were the subsequent non-negative terms that appeared in word cloud. There are additional words that described the activities that people doing throughout their lives, such as 'work' (3.76%) and 'working' (2.85%).

## Exploring gender differences: Does it matter from word cloud?

Since there are some notable differences in expression of depression between males and females. According to Cavanagh et al. (2016), depressed men reported alcohol and drugs misuse, poor impulse control, while depressed women more likely to report depressed mood, appetite, and sleep disturbance [54, 55]. There was also report which found that men have higher rate of anger [56]. Women tended to express their depression more accurately according to the criteria for depressive disorder, making it more difficult to detect same degree of depression in male patients. This section aims to explore whether gender differences influence the words produced in the word cloud from the LIME explanation.

| Females and Males | |
|---|---|
| **Word (Translation)** | **Sum of Explainable Weights** |
| ไม่ (not) | -5.865 |
| เศร้า (sad) | -2.144 |
| อารมณ์ (emotion) | -1.875 |
| ทำให้ (which make) | -1.703 |
| จน (until) | -1.358 |
| คงที่ (stable) | -1.305 |
| คิด (think) | -1.167 |
| นิ่ง (calm) | -1.092 |
| ฆ่าตัวตาย (suicide) | -1.045 |
| แต่ (but) | -0.898 |

| Females | |
|---|---|
| **Word (Translation)** | **Sum of Explainable Weights** |
| ไม่ (not) | -4.961 |
| ทำให้ (which make) | -1.703 |
| เศร้า (sad) | -1.446 |
| คิด (think) | -1.167 |
| เฉย (silently) | -0.927 |
| อารมณ์ (mood) | -0.823 |
| มาก (more) | -0.780 |
| เบื่อ (bore) | -0.759 |
| คงที่ (stable) | -0.725 |
| เหมือน (same) | -0.720 |

| Males | |
|---|---|
| **Word (Translation)** | **Sum of Explainable Weights** |
| จน (until) | -1.160 |
| นิ่ง (calm) | -1.092 |
| อารมณ์ (mood) | -1.052 |
| ฆ่าตัวตาย (suicide) | -1.045 |
| ไม่ (not) | -0.904 |
| แต่ (but) | -0.904 |
| แย่ (bad) | -0.779 |
| เศร้า (sad) | -0.698 |
| ถึงขั้น (to the point) | -0.693 |
| ก็ (subsequent) | -0.584 |

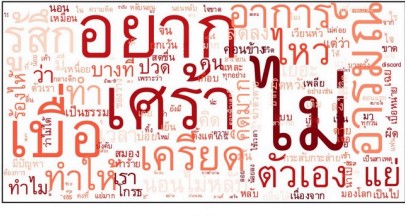
(a)

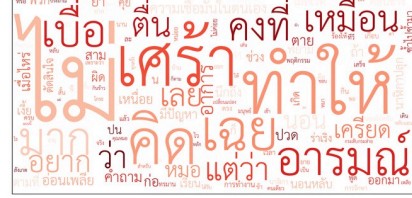
(b)

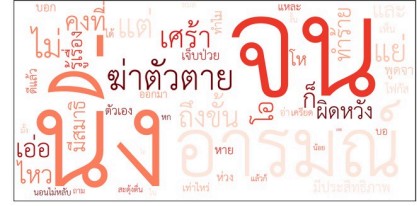
(c)

| Females and Males | |
|---|---|
| **Word (Translation)** | **Sum of Explainable Weights** |
| ช่วงนี้ (recently) | 27.157 |
| สบายดี (fine) | 14.432 |
| อย่างไรบ้าง (how) | 11.717 |
| ปกติ (normally) | 7.662 |
| คุณ (you) | 4.886 |
| กังวล (worry) | 4.071 |
| บ้าง (any) | 4.011 |
| ทำงาน (working) | 3.270 |
| ดี (good) | 3.058 |
| สบาย (comfortable) | 2.901 |

| Females - | |
|---|---|
| **Word (Translation)** | **Sum of Explainable Weights** |
| ช่วงนี้ (recently) | 31.700 |
| สบายดี (fine) | 17.140 |
| อย่างไรบ้าง (how) | 11.843 |
| ปกติ (normally) | 9.039 |
| คุณ (you) | 4.954 |
| ทำงาน (working) | 4.464 |
| กังวล (worry) | 4.071 |
| บ้าง (any) | 4.011 |
| สบาย (comfortable) | 3.251 |
| ดี (good) | 3.249 |

| Males | |
|---|---|
| **Word (Translation)** | **Sum of Explainable Weights** |
| ช่วงนี้ (recently) | 4.543 |
| สบายดี (fine) | 2.708 |
| ปกติ (normally) | 1.377 |
| งาน (work) | 1.259 |
| ทำงาน (working) | 1.195 |
| ตามปกติ (as usual) | 0.862 |
| น่าเป็นห่วง (worrisome) | 0.582 |
| เมนู (menu) | 0.458 |
| ให้ (give) | 0.443 |
| กด (press) | 0.428 |

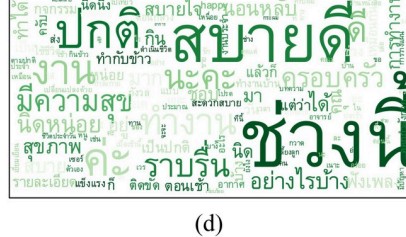
(d)

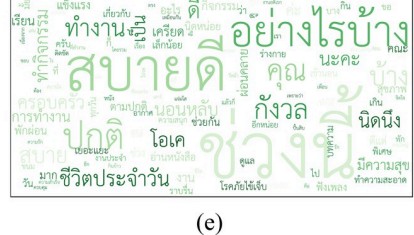
(e)

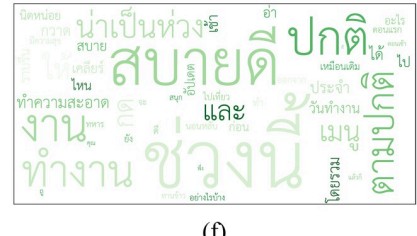
(f)

**Fig 3. Word cloud of the responses to the question "How are you these days?" from six different groups: a) depressed females and males, b) depressed females, c) depressed males, d) normal control females and male, e) normal control females, and f) normal control males.**

Based on our collected data, which show a prevalence of female and male depression with a ratio of 3:1 and were distributed evenly across severity (as seen in Table 2), the word clouds of $Q_1$ in Fig 3 produced from depressed females and males (Fig 3b) and 3c)), illustrated some shared words, such as 'not', 'sad', and 'mood', although in different orders. Similarly, a comparison of the word clouds produced from the normal control females and males (Fig 3e) and

[3f)](), reveals shared words such as 'recently', 'fine', 'normally', and 'working', but with the same orders. This suggests that the model tries to classify based on the common set of words even when there are some words that only occurred on each side of the gender.

## Error analysis of model misclassification

From Table 3, there were 33 support cases for the depression class in $Q_1$, indicating that 8 of 40 participants were misclassified. Among those misclassified, six were female and two were male. The sample response (English translation) from a misclassified male (Age = 63, Thai HAM-D Score = 15, Thai PHQ-9 Score = 16) was as follows:

> "Older people want to know why they were born and for what reason. But it can be concluded that, those who live must lead their own way of life and live happily as a normal human being should receive and make it the most useful for humanity in this world."

Upon carefully consideration, it was found that there were no distinct words in the word clouds produced from $Q_1$ in Fig 3 that indicated a feeling of depression. Furthermore, there were no words or phrases that indicated angry emotions, which are typically associated with male depression. More interestingly, out of three participants with comorbidity of bipolar and depression, two were female and one was male. Both female participants were misclassified by the model from $Q_1$, suggesting that the model may not be able to capture the text transcripts produced from psychiatric symptoms other than depression. Further investigation is needed to better understand the underlying factors behind the misclassification of the model.

# Conclusion & discussion
## Conclusion

Even when using a small amount of data to fine-tune a large T-PTLM model, such as the XLM-RoBERTa model, the model has a tendency to perform well in classification tasks. According to the findings of this study, using only text transcriptions from 40 depression participants and 40 normal control participants with K×L cross-validation techniques (all data points are treated as hold-out testing datasets) can achieve a classification sensitivity of 82.50% and a specificity of 85.00% by using only one question of "How are you these days?". The results can be improved by using an averaging ensemble of the first three questions ($Q_1 - Q_3$), which have a sensitivity of 87.50% and a specificity of 92.50%.

Based on the most accurate results from the first three questions, eight out of eighty participants were misclassified (five from depression and three from normal control). When analyzing the misclassifications in the depression group, we found that only one participant was misclassified on all three questions, while the rest were misclassified on two of the three. Examining the text transcript of the speech response from the depressed participant who was misclassified for all three questions, we found that the responses to $Q_1 - Q_3$ were "I'm fine, both mentally and physically, no problem," "I'm getting better," and "[Blank]" (no response), respectively. These responses were more likely to have come from normal control participants, as they were non-negative in tone and very brief. Although this participant was diagnosed with depression and received a Thai HAM-D score indicating mild severity (14 points), their Thai PHQ-9 score of 3 points was contradictory, as a score of 3 does not meet the diagnostic threshold for depression. This discrepancy could be due to patients falsely responding to the Thai version of the PHQ-9 in order to deny having any depressive symptoms.

When considering the LIME analysis, most of word clouds built from weight of the depression class have a preponderance towards a negative word than a neutral or positive one. This

was in contrast to normal control, where most of word clouds consisted of neutral or positive words. Some words are in conjunction with those found on the DSM-5 criteria for depression classification, such as the $1^{st}$ criterion of "Depressed mood most of the day, nearly everyday.", which can describe by the word 'sad' in the $2^{nd}$ rank in the word cloud or word 'mood' in the $3^{rd}$ rank in the word cloud from $Q_1$ in Fig 3a), or the $2^{nd}$ criterion of "Markedly diminished interest or pleasure in all, or almost all, activities most of the day, nearly every day.", which can describe by the word 'not' in the $1^{st}$ rank in the word cloud. Most importantly, the last criterion of "Recurrent thoughts of death, recurrent suicidal ideation without a specific plan, or a suicide attempt or a specific plan for committing suicide." can also describe by the word 'suicide' in the $9^{th}$ rank in the word cloud [45].

## Discussion

The methodology of this study was similar to previously reported studies in the literature, such as the work of Senn et al. (2022) [29], which used clinical transcripts from an interview with twelve questions [57] from 189 participants with the PHQ-8 as a support diagnosis. The RoBERTa model achieved an F1-score of 0.92 for classifying depression with only one question: "What are you most proud of?". When all the questions were used, the F1-score of the RoBERTa model decreased to 0.58 ± 0.22 (mean ± S.D.) [29]. Another work by Dai et al. (2021), also used the RoBERTa model for transfer learning with 500 discharge summaries from an electronic health record (EHR) system, and the F1-score was as high as 0.83 [28]. These experiments demonstrated that transfer learning towards a RoBERTa like model can also create a robust model even with a limited set of data. This work focused on leveraging the original Thai language, which is an uncommon domestic language, to create a multi-lingual support system with the XLM-RoBERTa model. The results revealed that the accuracy of the model was up to 90.00%, with only three questions. Although the model is based on the Thai language, the context of the speech responses to the questions was found to be generalizable to all practitioners, as shown in Fig 2. Further research and investigation into cross-language studies of this paradigm could be beneficial.

The words produced from our LIME explanation in word cloud are in line with previous published articles. Ghosh et al. (2021) used a traditional features-based method with LSTM to train a model from approximately 4.2 M tweets and achieved 87.14% accuracy. Their study noted that words such as 'depression', 'stress', and 'sad', as well as attributes such as 'work-pressure', 'divorce', and 'break-up' tended to have a positive correlation to depression [25]. This result was in alignment with the dataset collected by Wani et al. (2022), which showed that the most frequent depressed words from Facebook comments were 'fake', 'suicide', 'die', and 'sadness', while tweets were 'stress', 'life', 'failure', and YouTube comments were 'trauma', 'worst', and 'death' [24]. However, the other studies did not focus on the model's clinical implications [22, 23, 26, 28–31].

Although Ghosh et al. (2021) attempted to explain the model, their study focused on the weights directly which have some limitations. Since weights in a model are typically calculated from many input variables and interactions between variables, it can be difficult to understand the individual contributions of each factor to the overall result [40]. In comparison, our work utilized LIME explanations, which use various methods such as sampling, data perturbation, and feature importance to determine each feature's importance. This comprehensive method helps to better understand the importance of each model feature, as well as how changes to the data can affect the model output. Additionally, the analysis can identify any potential bias in the model, which is not possible when simply exploring the model weight directly [40].

Even though the performance of the classification model and explanation model yielded promising results, the study still produced some false hypothesis. This was due to the fact that the aim of the research paper was to create a set of twelve questions of Thai depression assessment tasks, which comprise of six questions for reviews of symptom ($Q_1 - Q_6$) and six questions for describing emotional experiences ($Q_7 - Q_{12}$). However, only the first six questions provided promising results. It is due to the categories of questions describing a daily life activity, while the rest of the questions were affected by polarity of emotion, such as $Q_9$ "Talk about an event that impresses you." and $Q_{11}$ "Talk about what makes you feel afraid.", which cannot differentiate between the depression group and the normal control. This scenario is similar to the one previously studied by Senn et al. (2022), where the use of a single question produced promising results [29]. However, when all questions were considered together, the results significantly dropped from an F1-score of 0.93 to 0.62, indicating that some questions were unnecessary for the model.

When examining the responses to the six questions related to describing emotional experiences ($Q_7 - Q_{12}$), it become evident that most responses were influenced by individual experiences. For example, a depressed patient responded that their life goal was to make their sibling happy ($Q_7$), and the event that impressed them the most was their sibling's ability to communicate with others ($Q_9$). Additionally, the patient's greatest fear was a family member's illness ($Q_{11}$). In contrast, a normal control participant responded that their life goal was to take care of their family, buy a house, and a car ($Q_7$), and the event that impressed them was being able to live with their family ($Q_9$). Finally, their greatest fear was the darkness ($Q_{11}$). Comparing these two examples demonstrated that questions about emotional experiences were based on personal experiences and may not be easily distinguished from one another. However, there are some data-related aspects to consider. For instance, in response to the question ($Q_9$),"Describe what makes you feel afraid," the majority of depressed individuals expressed a fear of losing a loved one, while the majority of normal control expressed a fear of the dark, ghosts, or animals such as lizards and cockroaches.

We are also mindful of the gender differences in terms of communicating and describing feelings of depression. Previously reported literature has shown that there is a gender difference in self-reporting symptom of depression in males and higher rates of suicide. Potential reasons behind this include gender differences in coping style, such as relative emotional inexpressiveness, non-help seeking behavior, as opposed to the reported feminine coping styles such as help-seeking and emotional expressiveness which can lead to major differences when filling out the questionnaire [58, 59]. There might be social expectations and stigma attached for males, associated with shame when expressing emotion or depression leading to an underdiagnosis and under-treatment of mild-moderate male depression and a higher suicide reporting in males [60]. Moreover, depressive symptoms in males usually including irritability, aggression, substance abuse, and risky behaviors [61]. Unfortunately, our collected dataset consisted of only 19 male data, which was insufficient to create a reliable model. This highlights the need for further research and questionnaire development to identify and screen for nuanced variations in depression reporting among different genders.

This study has already investigated the possibility of utilizing a very large T-PTLM to fine-tune the process of developing a robust classification model and to interpret the results. For other strength in this study, all questions in Thai Depression Assessment Tasks were adopted from normal routine in clinical practice, making it easy for patients to understand and user-friendly as an application. In addition, this study used Thai PHQ-9 for evaluation, which is a reliable tool and one of the most used questionnaire for depression screening. We also used Thai HAM-D, another questionnaire used to provide an indication of depression. Combination of these two evaluation tools helped increasing in accuracy of data collection and

reliability of data analysis. However, it should be noted that speech-to-text transcription alone may not be sufficient for screening other psychiatric conditions, such as bipolar type II or schizophrenia, as shown in the study by Dai et al. [28]. After taking into account all of these factors, it was concluded that speech-to-text transcription can be used for screening depression, but may not be effective for other psychiatric conditions or conditions that involve overlapping depressive symptoms.

For future direction, we plan to expand our study to investigate the relationship between normal control and depression in patients with other psychiatric disorders, such as bipolar disorder and psychotic disorder. This is because those with multiple psychological comorbidities might have variations in the presentation of depression, which will have implications for detection and diagnosis. Moreover, we plan to utilize data from various modalities to strengthen the accuracy and reliability of our models. To further improve our models, we are also aiming to employ optimization techniques [62, 63]. Furthermore, we are hoping to use our models for clinical studies, which will allow us to gain better insights into the development of these psychiatric illnesses. Developing an application which can detect depression will help with insufficient health care workers' problem, and this would also be beneficial to improve health care service accessibility.

## Supporting information

**S1 Data.**
(XLSX)

**S2 Data.**
(XLSX)

**S3 Data.**
(XLSX)

**S1 Text.**
(TXT)

## Author Contributions

**Conceptualization:** Adirek Munthuli, Chutamanee Onsuwan, Kankamol Jaisin, Keerati Pattanaseri, Juthawadee Lortrakul, Charturong Tantibundhit.

**Data curation:** Adirek Munthuli.

**Formal analysis:** Adirek Munthuli, Nittayapa Klangpornkun.

**Investigation:** Adirek Munthuli, Nittayapa Klangpornkun, Chutamanee Onsuwan, Kankamol Jaisin, Keerati Pattanaseri, Juthawadee Lortrakul, Charturong Tantibundhit.

**Methodology:** Adirek Munthuli, Nittayapa Klangpornkun, Chutamanee Onsuwan, Kankamol Jaisin, Keerati Pattanaseri, Juthawadee Lortrakul, Charturong Tantibundhit.

**Project administration:** Chutamanee Onsuwan, Kankamol Jaisin, Keerati Pattanaseri, Juthawadee Lortrakul, Charturong Tantibundhit.

**Software:** Phongphan Phienphanich.

**Supervision:** Chutamanee Onsuwan, Kankamol Jaisin, Keerati Pattanaseri, Juthawadee Lortrakul, Charturong Tantibundhit.

**Validation:** Adirek Munthuli.

**Visualization:** Adirek Munthuli.

**Writing – original draft:** Adirek Munthuli.

**Writing – review & editing:** Pakinee Pooprasert, Chutamanee Onsuwan, Kankamol Jaisin, Keerati Pattanaseri, Juthawadee Lortrakul, Charturong Tantibundhit.

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
