## [Decision Letter · Decision Letter 0]

6 Jan 2023

PONE-D-22-34559Classification and Analysis of Text Transcription from Thai Depression Assessment Tasks among Patients with DepressionPLOS ONE

Dear Dr. Lortrakul,

Thank you for submitting your manuscript to PLOS ONE. After careful consideration, we feel that it has merit but does not fully meet PLOS ONE’s publication criteria as it currently stands. Therefore, we invite you to submit a revised version of the manuscript that addresses the points raised during the review process.

We look forward to receiving your revised manuscript.

Kind regards,

Sathishkumar V E

Academic Editor

PLOS ONE

Journal Requirements:

Reviewers' comments:

Reviewer's Responses to Questions

**Comments to the Author**

1. Is the manuscript technically sound, and do the data support the conclusions?

Reviewer #1: Yes

Reviewer #2: Yes

2. Has the statistical analysis been performed appropriately and rigorously? 

Reviewer #1: No

Reviewer #2: I Don't Know

3. Have the authors made all data underlying the findings in their manuscript fully available?

Reviewer #1: Yes

Reviewer #2: Yes

4. Is the manuscript presented in an intelligible fashion and written in standard English?

Reviewer #1: Yes

Reviewer #2: Yes

5. Review Comments to the Author

Reviewer #1: This paper has 40 controls and 40 depressed patients for predictability of the text they use for depression.

Here are the problems:

1- Men and women do not experience depression similarly. This needs to be core to this paper. All analyses should be by gender too. Additional tables and figures are needed.

2- Men show anger and women cry, when depressed. That means, text should be gender specific, at least in part.

3- Women start using care sooner than men when depressed. That means, men have more severe depression in clinics, because they use care too late. This has implications for this study.

4- This paper does not have genera international audience. Thus, I am not sure if this is a good fit for the PLOS One.

Reviewer #2: Comments to Author:

This paper is presented well, and it needs some enhancements, which are as follows.

The contributions of this work are not clear. I went through the abstract and introduction, I did not get the main contribution of this work I suggest the authors spend significant efforts to enhance the main work in this research

The research findings and contribution need to be stated clearly. As well as the obtained results in this paper.

The introduction, e.g., should lead the way throughout the paper. In addition, the benefits coming from this research should be made clearer in the introduction and throughout the paper. I believe, this section needs significant efforts to make things and contributions clearer and flow of the contents.

The related work section is clear, and it needs a discussion for the studied papers.

The problem scenario is not clear.

The mathmatical notations need more focus and enhancement.

The used figures in this paper are not clear enough

I need to see the main procedure of the proposed method

Some works are worthy of mentioning in this paper:

Prairie dog optimization algorithm

Gazelle Optimization Algorithm: A novel nature-inspired metaheuristic optimizer

Dwarf Mongoose Optimization Algorithm

Ebola Optimization Search Algorithm: A new nature-inspired metaheuristic algorithm

Reptile Search Algorithm (RSA): A nature-inspired meta-heuristic optimizer

The arithmetic optimization algorithm

Aquila Optimizer: A novel meta-heuristic optimization algorithm

Furthermore, the evaluation metrics should be briefly described in the experimental section. Moreover, add further details on how simulations were conducted. Similarly, system and resource characteristics could be added to Tables for clarity.

The conclusion section also needs significant revisions. It should briefly describe the study's findings and some more directions for further research. The authors should describe academic implications, major findings, shortcomings, and directions for future research in the conclusion section.

Can you compare the proposed method with other published method to solve the same case

6. PLOS authors have the option to publish the peer review history of their article (what does this mean?). If published, this will include your full peer review and any attached files.

Reviewer #1: No

Reviewer #2: No

<quillbot-extension-portal></quillbot-extension-portal>

---

## [Author Response · Author response to Decision Letter 0]

23 Feb 2023

Reviewer#1, Concern # 1: Men and women do not experience depression similarly. This needs to be core to this paper. All analyses should be by gender too. Additional tables and figures are needed.

Author response: 

Exploring Gender Differences: Does it Matter from Word Cloud?

Since there are some notable differences in expression of depression between males and females. According to Cavanagh et al. (2016), depressed men reported alcohol and drugs misuse, poor impulse control, while depressed women more likely to report depressed mood, appetite, and sleep disturbance [55, 56]. There was also report which found that men have higher rate of anger [57]. Women tended to express their depression more accurately according to the criteria for depressive disorder, making it more difficult to detect same degree of depression in male patients. This section aims to explore whether gender differences influence the words produced in the word cloud from the LIME explanation.

Based on our collected data, which show a prevalence of female and male depression with a ratio of 3:1 and were distributed evenly across severity (as seen in Table 2), the word clouds of Q1 in Fig. 3 produced from depressed females and males (Fig. 3b)) and (Fig. 3c)), illustrated some shared words, such as ‘not’, ‘sad’, and ‘mood’, although in different orders. Similarly, a comparison of the word clouds produced from the normal control females and males (Fig. 3e)) and (Fig. 3f)), reveals shared words such as ‘recently’, ‘fine’, ‘normally’, and ‘working’, but with the same orders. This suggests that the model tries to classify based on the common set of words even when there are some words that only occurred on each side of the gender.

Reviewer#1, Concern # 2: Men show anger and women cry, when depressed. That means, text should be gender specific, at least in part.

Author response: 

Error Analysis of Model Misclassification

From Table 3, there were 33 support cases for the depression class in Q1, indicating that 8 of 40 participants were misclassified. Among those misclassified, six were female and two were male. The sample response (English translation) from a misclassified male (Age=63, Thai HAM-D Score=15, Thai PHQ-9 Score=16) was as follows:

“Older people want to know why they were born and for what reason. But it can be concluded that, those who live must lead their own way of life and live happily as a normal human being should receive and make it the most useful for humanity in this world.”

Upon carefully consideration, it was found that there were no distinct words in the word clouds produced from Q1 in Fig. 3 that indicated a feeling of depression. Furthermore, there were no words or phrases that indicated angry emotions, which are typically associated with male depression. More interestingly, out of three participants with comorbidity of bipolar and depression, two were female and one was male. Both female participants were misclassified by the model from Q1, suggesting that the model may not be able to capture the text transcripts produced from psychiatric symptoms other than depression. Further investigation is needed to better understand the underlying factors behind the misclassification of the model.

Reviewer#1, Concern # 3: Women start using care sooner than men when depressed. That means, men have more severe depression in clinics, because they use care too late. This has implications for this study.

Author response: 

We are also mindful of the gender differences in terms of communicating and describing feelings of depression. Previously reported literature has shown that there is a gender difference in self-reporting symptom of depression in males and higher rates of suicide. Potential reasons behind this include gender differences in coping style, such as relative emotional inexpressiveness, non-help seeking behavior, as opposed to the reported feminine coping styles such as help-seeking and emotional expressiveness which can lead to major differences when filling out the questionnaire [59, 60]. There might be social expectations and stigma attached for males, associated with shame when expressing emotion or depression leading to an under-diagnosis and under-treatment of mild-moderate male depression and a higher suicide reporting in males [61]. Moreover, depressive symptoms in males usually including irritability, aggression, substance abuse, and risky behaviors [62]. Unfortunately, our collected dataset consisted of only 19 male data, which was insufficient to create a reliable model. This highlights the need for further research and questionnaire development to identify and screen for nuanced variations in depression reporting among different genders.

Reviewer#1, Concern # 4: This paper does not have genera international audience. Thus, I am not sure if this is a good fit for the PLOS One.

Author response: 

The methodology of this study was similar to previously reported studies in the literature, such as the work of Senn et al. (2022) [30], which used clinical transcripts from an interview with twelve questions [58] from 189 participants with the PHQ-8 as a support diagnosis. The RoBERTa model achieved an F1-score of 0.92 for classifying depression with only one question: “What are you most proud of?”. When all the questions were used, the F1-score of the RoBERTa model decreased to 0.58 ± 0.22 (mean ± S.D.) [30]. Another work by Dai et al. (2021), also used the RoBERTa model for transfer learning with 500 discharge summaries from an electronic health record (EHR) system, and the F1-score was as high as 0.83 [29]. These experiments demonstrated that transfer learning towards a RoBERTa like model can also create a robust model even with a limited set of data. This work focused on leveraging the original Thai language, which is an uncommon domestic language, to create a multi-lingual support system with the XLM-RoBERTa model. The results revealed that the accuracy of the model was up to 90.00%, with only three questions. Although the model is based on the Thai language, the context of the speech responses to the questions was found to be generalizable to all practitioners, as shown in Fig. 2. Further research and investigation into cross-language studies of this paradigm could be beneficial.

The words produced from our LIME explanation in word cloud are in line with previous published articles. Ghosh et al. (2021) used a traditional features-based method with LSTM to train a model from approximately 4.2 M tweets and achieved 87.14% accuracy. Their study noted that words such as ‘depression’, ‘stress’, and ‘sad’, as well as attributes such as ‘work-pressure’, ‘divorce’, and ‘break-up’ tended to have a positive correlation to depression [26]. This result was in alignment with the dataset collected by Wani et al. (2022), which showed that the most frequent depressed words from Facebook comments were ‘fake’, ‘suicide’, ‘die’, and ‘sadness’, while tweets were ‘stress’, ‘life’, ‘failure’, and YouTube comments were ‘trauma’, ‘worst’, and ’death’ [25]. However, the other studies did not focus on the model’s clinical implications [23, 24, 27, 29–32].

Reviewer#2, Concern # 1: The contributions of this work are not clear. I went through the abstract and introduction, I did not get the main contribution of this work I suggest the authors spend significant efforts to enhance the main work in this research. 

Author response: 

We modified the abstract and enhanced the main contributions of this research to be clearer. The modified abstract is shown below. In addition, we spent significant efforts to enhance contributions of this research in the introduction section described in the author response to Reviewer#2, Concern # 2.

Abstract

Depression is a serious mental health disorder that poses a major public health concern in Thailand and have a profound impact on individuals’ physical and mental health. In addition, the lack of number to mental health services and limited number of psychiatrists in Thailand make depression particularly challenging to diagnose and treat, leaving many individuals with the condition untreated. Recent studies have explored the use of natural language processing to enable access to the classification of depression, particularly with a trend toward transfer learning from pre-trained language model. In this study, we attempted to evaluate the effectiveness of using XLM-RoBERTa, a pre-trained multilingual language model supporting the Thai language, for the classification of depression from a limited set of text transcripts from speech responses. Twelve Thai depression assessment questions were developed to collect text transcripts of speech responses to be used with XLM-RoBERTa in transfer learning. The results of transfer learning with text transcription from speech responses of 80 participants (40 with depression and 40 normal control) showed that when only one question (Q1) of “How are you these days?” was used, the recall, precision, specificity, and accuracy were 82.5%, 84.65, 85.00, and 83.75%, respectively. When utilizing the first three questions from Thai depression assessment tasks (Q1 −Q3), the values increased to 87.50%, 92.11%, 92.50%, and 90.00%, respectively. The local interpretable model explanations were analyzed to determine which words contributed the most to the model’s word cloud visualization. Our findings were consistent with previously published literature and provide similar explanation for clinical settings. It was discovered that the classification model for individuals with depression relied heavily on negative terms such as ‘not,’ ‘sad,’, ‘mood’, ‘suicide’, ‘bad’, and ‘bore’ whereas normal control participants used neutral to positive terms such as ‘recently,’ ‘fine,’, ‘normally’, ‘work’, and ‘working’. The findings of the study suggest that screening for depression can be facilitated by eliciting just three questions from patients with depression, making the process more accessible and less time-consuming while reducing the already huge burden on healthcare workers.

Reviewer#2, Concern # 2: The research findings and contribution need to be stated clearly. As well as the obtained results in this paper.

Author response: 

Our four main research contributions include:

1. Fine-tuning XLM-ROBERTABASE [45] to text transcripts elicited from spoken responses to twelve questions of Thai depression assessment tasks [44].

2. Assessing the effectiveness of Thai depression assessment tasks for detecting depression from text responses in speech and outline a set of questions that yield the most accurate results in clinical settings.

3. Explaining a black-box model from fine-tuned XLM-RoBERTa using weight explained from LIME explanations [41]. 

4. Visualizing weights from LIME explanations [41] in a word cloud format and explaining their relevance to clinical settings.

Reviewer#2, Concern # 3: The introduction, e.g., should lead the way throughout the paper. In addition, the benefits coming from this research should be made clearer in the introduction and throughout the paper. I believe, this section needs significant efforts to make things and contributions clearer and flow of the contents.

Author response: This research has important clinical implications, as it demonstrates that a highly accurate model for classifying depression can be produced using a limited set of high-quality text transcripts from speech in clinical settings. This is particularly noteworthy for uncommon domestic languages such as Thai, where data availability is often a significant issue. Additionally, the explanations provided by LIME through the black-box model can be interpreted and understood in a clinical context.

Reviewer#2, Concern # 4: The related work section is clear, and it needs a discussion for the studied papers.

Author response: 

Introduction

In recent years, several studies have developed classification models from social media posts using vanilla fine-tuning of BERT and the model can successfully achieve up to 98% accuracy [26–28, 31, 32]. Instead of using social media posts, Dai et al. (2021) provided more examples of their work. They utilized transfer learning from four variants of the BERT model to identify depression in discharge summaries, and RoBERTa yielded the highest classification F1-score of 0.830 [29]. Another example was the work by Senn et al. (2022), which used transcripts of 12 clinical interview questions and a transfer learning approach with three variants of BERT and an ensemble of extra LSTM and attention layers. The results indicated that RoBERTa tended to perform better with basic architectures individually without ensemble. However, the best result was achieved using the BERT architecture with ensemble strategy of only one question, with an F1-score of 0.93. When using all the questionnaires, the F1-score dropped to 0.62 ± 0.12 (mean ± S.D.) [30].

Discussion

The methodology of this study was similar to previously reported studies in the literature, such as the work of Senn et al. (2022) [30], which used clinical transcripts from an interview with twelve questions [58] from 189 participants with the PHQ-8 as a support diagnosis. The RoBERTa model achieved an F1-score of 0.92 for classifying depression with only one question: “What are you most proud of?”. When all the questions were used, the F1-score of the RoBERTa model decreased to 0.58 ± 0.22 (mean ± S.D.) [30]. Another work by Dai et al. (2021), also used the RoBERTa model for transfer learning with 500 discharge summaries from an electronic health record (EHR) system, and the F1-score was as high as 0.83 [29]. These experiments demonstrated that transfer learning towards a RoBERTa like model can also create a robust model even with a limited set of data. This work focused on leveraging the original Thai language, which is an uncommon domestic language, to create a multi-lingual support system with the XLM-RoBERTa model. The results revealed that the accuracy of the model was up to 90.00%, with only three questions. Although the model is based on the Thai language, the context of the speech responses to the questions was found to be generalizable to all practitioners, as shown in Fig. 2. Further research and investigation into cross-language studies of this paradigm could be beneficial.

The words produced from our LIME explanation in word cloud are in line with previous published articles. Ghosh et al. (2021) used a traditional features-based method with LSTM to train a model from approximately 4.2 M tweets and achieved 87.14% accuracy. Their study noted that words such as ‘depression’, ‘stress’, and ‘sad’, as well as attributes such as ‘work-pressure’, ‘divorce’, and ‘break-up’ tended to have a positive correlation to depression [26]. This result was in alignment with the dataset collected by Wani et al. (2022), which showed that the most frequent depressed words from Facebook comments were ‘fake’, ‘suicide’, ‘die’, and ‘sadness’, while tweets were ‘stress’, ‘life’, ‘failure’, and YouTube comments were ‘trauma’, ‘worst’, and ’death’ [25]. However, the other studies did not focus on the model’s clinical implications [23, 24, 27, 29–32].

Although Ghosh et al. (2021) attempted to explain the model, their study focused on the weights directly which have some limitations. Since weights in a model are typically calculated from many input variables and interactions between variables, it can be difficult to understand the individual contributions of each factor to the overall result [41]. In comparison, our work utilized LIME explanations, which use various methods such as sampling, data perturbation, and feature importance to determine each feature’s importance. This comprehensive method helps to better understand the importance of each model feature, as well as how changes to the data can affect the model output. Additionally, the analysis can identify any potential bias in the model, which is not possible when simply exploring the model weight directly [41].

Reviewer#2, Concern # 5: The problem scenario is not clear.

Author response: 

The goal of this study is to examine the potential of using a limited-set of text transcripts from speech responses to fine-tune a T-PTLM to create a promising classification model [39]. The main issues addressed in this research are (1) the scarcity of textual transcripts from speech collected in clinical settings, (2) how to develop a highly accurate model using T-PTLM with a limited dataset, and (3) how the results can be applied and trusted in clinical settings. Figure 1 illustrates the process from data collection, data preparation, and fine-tuning of the model. The details of this process are provided in the following subsections.

Reviewer#2, Concern # 6: The mathematical notations need more focus and enhancement.

Author response: 

Model Explanation and Visualization

The model will later be explained by LIME explanation [41] to determine which words contribute the most for depression screening. The idea of LIME explanation was to randomly make an absence of words from text transcripts (perturb) and to create white-box model that resembles the black-box model (fine-tuned XLM-RoBERTa model) and minimizes the following equations:

 {\\mathrm{arg\\thinspmin}}_{g\\in G}\\mathcal{L}\\left(f,g,\\pi_x\\right)+\\Omega\\left(g\\right) (5)

 \\pi_x\\left(z\\right)=e^{-\\frac{D\\left(x,z\\right)^2}{\\sigma^2}} (6)

where x is the instance to be explained, z is the perturbed instance, f is the black-box model to be explained, G is the hypothesis space of white-box models, \\mathcal{L}\\left(f,g,\\pi_x\\right) is the deviation between the black-box model (f) and the white-box model (g), as weight measured on a perturbed version of the text transcripts x, represented by \\pi_x\\left(z\\right), \\Omega\\left(g\\right) is a regularization term that controls the complexity of the interpretable model (g), D\\left(x,z\\right) is a cosine similarity between x and z, and \\sigma is kernel width. 

The goal of LIME is to find the white-box model (g) that is closest to the black-box model (f) and at the same time has the lowest complexity, as determined by \\Omega\\left(g\\right). The result of LIME is an explanation of the predictions of the black-box model (f) for the instance x. In this work, we implement a LIME analysis using the ELI5 library [54]. With ELI5, we can obtain an interpretable description of the model’s predictions along with the individual contributions of each word (\\pi_x\\left(z\\right)). 

Since binary classification models produce four different weights in LIME explanations, namely the weight of the true positive class (depression predicted as depression), the weight of the false negative class (depression predicted as normal control), the weight of the false positive class (normal control predicted as depression), and the weight of the true negative class (normal control predicted as normal control), the relationship between words and the prediction of depression or normal control will be examined by summarizing the LIME weights from the true positive and true negative classes as a word cloud. To calculate this, we summed all the weights that had a positive effect on each class and normalized them to a scale between 0 and 1.

Reviewer#2, Concern # 7: The used figures in this paper are not clear enough.

Author response: 

Fig 1. A flow diagram of the article’s process including a) data collection and speech elicitation, which are then transformed into text transcriptions; b) text pre-processing and normalization, followed by data preparation for cross-validation in transfer learning to the model for each question to compute evaluation matrices; c) training of the model for LIME explanation and utilization of the weights from the explanation for creating word clouds for both the normal control group and the depression group.

Fig 2. Samples of transcription of speech response from participants with original language (Thai) and English translation.

Fig 3. Word cloud of the responses to the question “How are you these days?” from six different groups: a) depressed females and males, b) depressed females, c) depressed males, d) normal control females and male, e) normal control females, and f) normal control males.

Reviewer#2, Concern # 8: I need to see the main procedure of the proposed method.

Author response: 

Fig 1. A flow diagram of the article’s process including a) data collection and speech elicitation, which are then transformed into text transcriptions; b) text pre-processing and normalization, followed by data preparation for cross-validation in transfer learning to the model for each question to compute evaluation matrices; c) training of the model for LIME explanation and utilization of the weights from the explanation for creating word clouds for both the normal control group and the depression group.

Reviewer#2, Concern # 9: Some works are worthy of mentioning in this paper: Prairie dog optimization algorithm, Gazelle Optimization Algorithm: A novel nature-inspired metaheuristic optimizer, Dwarf Mongoose Optimization Algorithm, Ebola Optimization Search Algorithm: A new nature-inspired metaheuristic algorithm, Reptile Search Algorithm (RSA): A nature-inspired meta-heuristic optimizer, The arithmetic optimization algorithm, Aquila Optimizer: A novel meta-heuristic optimization algorithm.

Author action: 

Author response: 

For future direction, we plan to expand our study to investigate the relationship between normal control and depression in patients with other psychiatric disorders, such as bipolar disorder and psychotic disorder. This is because those with multiple psychological comorbidities might have variations in the presentation of depression, which will have implications for detection and diagnosis. Moreover, we plan to utilize data from various modalities to strengthen the accuracy and reliability of our models. To further improve our models, we are also aiming to employ optimization techniques [63,64]. Furthermore, we are hoping to use our models for clinical studies, which will allow us to gain better insights into the development of these psychiatric illnesses. Developing an application which can detect depression will help with insufficient health care workers’ problem, and this would also be beneficial to improve health care service accessibility.

Reviewer#2, Concern # 10: The evaluation metrics should be briefly described in the experimental section. Moreover, add further details on how simulations were conducted. Similarly, system and resource characteristics could be added to Tables for clarity.

Author response: 

Evaluation Matrices

Our goal is to maximize recall (sensitivity), precision, and specificity of the model, as these matrices are widely used for clinical applications. Additionally, accuracy is reported to provide a holistic view of performance. Thus, in this work, the performance of a model is evaluated based on a variety of metrics, including recall, precision, specificity, and accuracy, using the following equations [53]:

Recall = TP/(TP+FN)

Precision = TP/(TP+FP)

Specificity = TN/(TN+FP)

Accuracy = (TP+TN)/(TP+TN+TP+FN)

where TP is true positive (depression predicted as depression), TN is true negative (normal control predicted as normal control), FP is false positive (normal control predicted as depression), and FN is false negative (depression predicted as normal control).

Implementation of Fine-tuning XLM-RoBERTa Model

The XLM-RoBERTa model was implemented for fine-tuning the data using the Keras library [50] on an NVIDIA RTX A6000 with 128 GB of RAM and 1 TB of SSD. The model was trained using a triangular learning rate policy [51] with a reduction factor of 2 and a reduction on plateau of 4. Other hyperparameters included a maximum of 50 epochs of training, a batch size of 8, and a learning rate of 1e-5. Furthermore, the model for each question was trained individually to examine the potential of the questions for the performance of the classification model.

Reviewer#2, Concern # 11: The conclusion section also needs significant revisions. It should briefly describe the study's findings and some more directions for further research. The authors should describe academic implications, major findings, shortcomings, and directions for future research in the conclusion section.

Author response: 

When consider the parts of text transcriptions that contributes to which one of normal control or depression When considering conducting the LIME analysis, most of word cloud built from weight of the depression class has a preponderance towards a negative word than a neutral or positive one. This was in contrasted to normal controls, where most of word cloud consisted of neutral or positive words. Some of words are in conjunction with those found on the DSM-5 criteria for depression classification, such as the 1st criteria of “Depressed mood most of the day, nearly everyday.”, which can be described by the word ‘Sad’ in 2nd rank of word cloud or word ‘mood’ in 3rd rank of word cloud from Q1 in Fig. 3a), or the 2nd criteria of “Markedly diminished interest or pleasure in all, or almost all, activities most of the day, nearly every day.”, which can be described by the word ‘negate’ in 1st rank of word cloud. Most importantly, the last criteria of “Recurrent thoughts of death, recurrent suicidal ideation without a specific plan, or a suicide attempt or a specific plan for committing suicide.” can also described be by the word ‘suicide’ in 9th rank of word cloud [46].

Reviewer#2, Concern # 12: Can you compare the proposed method with other published method to solve the same case.

Author response: 

The methodology of this study was similar to previously reported studies in the literature, such as the work of Senn et al. (2022) [30], which used clinical transcripts from an interview with twelve questions [58] from 189 participants with the PHQ-8 as a support diagnosis. The RoBERTa model achieved an F1-score of 0.92 for classifying depression with only one question: “What are you most proud of?”. When all the questions were used, the F1-score of the RoBERTa model decreased to 0.58 ± 0.22 (mean ± S.D.) [30]. Another work by Dai et al. (2021), also used the RoBERTa model for transfer learning with 500 discharge summaries from an electronic health record (EHR) system, and the F1-score was as high as 0.83 [29]. These experiments demonstrated that transfer learning towards a RoBERTa like model can also create a robust model even with a limited set of data. This work focused on leveraging the original Thai language, which is an uncommon domestic language, to create a multi-lingual support system with the XLM-RoBERTa model. The results revealed that the accuracy of the model was up to 90.00%, with only three questions. Although the model is based on the Thai language, the context of the speech responses to the questions was found to be generalizable to all practitioners, as shown in Fig. 2. Further research and investigation into cross-language studies of this paradigm could be beneficial.

The words produced from our LIME explanation in word cloud are in line with previous published articles. Ghosh et al. (2021) used a traditional features-based method with LSTM to train a model from approximately 4.2 M tweets and achieved 87.14% accuracy. Their study noted that words such as ‘depression’, ‘stress’, and ‘sad’, as well as attributes such as ‘work-pressure’, ‘divorce’, and ‘break-up’ tended to have a positive correlation to depression [26]. This result was in alignment with the dataset collected by Wani et al. (2022), which showed that the most frequent depressed words from Facebook comments were ‘fake’, ‘suicide’, ‘die’, and ‘sadness’, while tweets were ‘stress’, ‘life’, ‘failure’, and YouTube comments were ‘trauma’, ‘worst’, and ’death’ [25]. However, the other studies did not focus on the model’s clinical implications [23, 24, 27, 29–32].

Although Ghosh et al. (2021) attempted to explain the model, their study focused on the weights directly which have some limitations. Since weights in a model are typically calculated from many input variables and interactions between variables, it can be difficult to understand the individual contributions of each factor to the overall result [41]. In comparison, our work utilized LIME explanations, which use various methods such as sampling, data perturbation, and feature importance to determine each feature’s importance. This comprehensive method helps to better understand the importance of each model feature, as well as how changes to the data can affect the model output. Additionally, the analysis can identify any potential bias in the model, which is not possible when simply exploring the model weight directly [41].

---

## [Editor Report · Decision Letter 1]

2 Mar 2023

Classification and Analysis of Text Transcription from Thai Depression Assessment Tasks among Patients with Depression

PONE-D-22-34559R1

Dear Dr. Lortrakul,

We’re pleased to inform you that your manuscript has been judged scientifically suitable for publication and will be formally accepted for publication once it meets all outstanding technical requirements.

Kind regards,

Sathishkumar V E

Academic Editor

PLOS ONE
---

## [Editor Report · Acceptance letter]

21 Mar 2023

PONE-D-22-34559R1 

Classification and Analysis of Text Transcription from Thai Depression Assessment Tasks among Patients with Depression 

Dear Dr. Lortrakul:

I'm pleased to inform you that your manuscript has been deemed suitable for publication in PLOS ONE. Congratulations! Your manuscript is now with our production department. 

Kind regards, 

on behalf of

Dr. Sathishkumar V E 

Academic Editor

PLOS ONE